# Neuro-symbolic Learning of Lifted Action Models from Visual Traces

**Primary Keywords:** *(2) Learning;*

### Abstract

Model-based planners rely on action models to describe available actions in terms of their preconditions and effects. Nonetheless, manually encoding such models is challenging, especially in complex domains. Numerous methods have been proposed to learn action models from examples of plan execution traces. However, high-level information, such as state labels within traces, is often unavailable and needs to be inferred indirectly from raw observations. In this paper, we aim to learn lifted action models from visual traces — sequences of image-action pairs depicting discrete successive trace steps. We present ROSAME, a differentiable neu**RO**-**S**ymbolic **A**ction **M**odel l**E**arner that infers action models from traces consisting of probabilistic state predictions and actions. By combining ROSAME with a deep learning computer vision model, we create an end-to-end framework that jointly learns state predictions from images and infers symbolic action models. Experimental results demonstrate that our method succeeds in both tasks, using different visual state representations, with the learned action models often matching or even surpassing those created by humans.

## 1 Introduction

AI planning seeks to automatically identify an optimal course of action for an agent to achieve a goal within its environment. Planning algorithms typically rely on a planning domain model as input. The most critical component in a planning domain model is the action model, which describes the preconditions and effects of each action, enabling planners to reason about available actions and infer their outcomes. However, obtaining such action models can be challenging. Traditionally, they are often handcrafted by human experts, making it expensive, time-consuming, and error-prone. Acquiring action models from observational data would be much more cost-effective and reliable. Many proposals for this task assume fully observable states and actions (Pasula, Zettlemoyer, and Kaelbling 2007; Jiménez, Fernández, and Borrajo 2008; Rodrigues et al. 2012; Lamanna et al. 2021). The problem is that state and action labels may not always be available. Labelling numerous propositions in each state is particularly costly, leading to many other attempts at reducing reliance on state observability (McCluskey, Richardson, and Simpson 2002; Yang, Wu, and Jiang 2007; McCluskey et al. 2010; Zhuo et al. 2010; Zhuo, Muñoz-Avila, and Yang 2011; Cresswell and Gregory 2011; Cresswell, McCluskey, and West 2013; Zhuo and Kambhampati 2013; Aineto, Jiménez Celorrio, and Onaindia 2019). However, such reduction often comes at the expense of other aspects, including but not limited to completeness, quality, and readability of the learned models. Recent advancements in deep learning allow predicting states and actions from raw observations using neural networks, potentially striking a better balance between data collection cost and learning outcome quality. Nevertheless, learning action models typically involve symbolic logical inference, which is generally non-differentiable. How to effectively combine such symbolic inference with deep learning remains an open problem.

Encouraged by the emergence of neuro-symbolic techniques (Wang et al. 2019; Pogancic et al. 2020; Ahmed et al. 2022), we aim to meet this challenge with a neuro-symbolic model. An ideal scenario would be to learn action models directly from video demonstrations of executed plans without annotated supervision. In this paper, we take the first step toward this goal by addressing the simpler problem of learning action models from visual traces, where we only observe images depicting the states, and not the states labels directly, as shown in Fig. 1. By introducing a differentiable relaxation of the rules governing action models, we can integrate such a neuro-symbolic model with a deep learning computer vision model applied to visual observations, thereby formulating an end-to-end method to jointly learn human-readable, lifted action models and state predictors from image sequences. We conduct experiments in several planning domains using two different types of visual state representations. The action models learned by our method closely resemble, and sometimes improve on, those written by humans.

## 2 Related Work

One of the barriers to learning action models is acquiring a sufficient amount of supervised data. Obtaining fully-observed state (proposition) labels is exceptionally expensive. Some earlier works, such as LOCM (Cresswell, McCluskey, and West 2009; Cresswell and Gregory 2011; Cresswell, McCluskey, and West 2013) and Opmaker (McCluskey, Richardson, and Simpson 2002; McCluskey et al. 2010), operate under the assumption that there is no direct observation of states; they only take action sequences as input. These methods infer states and predicates using heuristic rules but provide no guarantee of completeness, soundness, or the quality of the learned models.

A few more recent works have tackled, in a principled way, the problems of simultaneously creating the symbols required to represent the domain and learning action models based on the created symbols (Konidaris, Kaelbling, and Lozano-Pérez 2014, 2018; Bonet and Geffner 2020; Rodriguez et al. 2021). In comparison, while we assume that the symbols are given, our task is to jointly learn the action models and ground the symbols from observed state images. Additionally, while these methods bypass the cost of state labelling, they require structured environment descriptions (such as state-space graphs or a semi-Markov decision process) as input, which demands a significant amount of technical expertise to construct. In contrast, our methods rely solely on observing demonstrations of planning tasks.

Another approach to reducing reliance on state labelling is to provide alternative information that is easier to access, such as state images. A significant contribution in this area is Latplan (Asai and Fukunaga 2018; Asai and Kajino 2019; Asai and Muise 2020; Asai et al. 2022), an unsupervised neuro-symbolic model, based on an auto-encoder framework that exclusively utilizes state images to recover action models. While both Latplan and our approach apply neuro-symbolic methods to state images, there are significant distinctions between the two works, leading to complementary strengths and weaknesses. As an unsupervised framework, Latplan requires no ground truth labelling and operates within a latent space, which grants it the flexibility to handle domains that are otherwise challenging to express. However, the lack of interpretability of the latent model poses challenges in model verification and evaluation. Translating actions from the latent space into actions that can be physically executed in the real world is also challenging. In contrast, we aim to learn human-readable models, at the cost of providing minimal additional information, in the form of the signature of the predicates and action symbols of the model sought. Moreover, we produce first-order models, whereas the models learned by Latplan are propositional. An extension of Latplan was able to learn first-order representations for states (Asai 2019), but this has yet to be generalized to action models. Liberman, Bonet, and Geffner (2022) presented a formulation to learn first-order representations from parsed images. However, they did not discuss integrating the deep learning image parsing model with their formulation to create an end-to-end method.

## 3 Preliminaries

Here we define the planning models we consider and introduce our notations. We assume that the reader is familiar with first-order logic, including with the concept of substitution. We write $\varphi[\sigma]$ for the application of substitution $\sigma$ to a first-order logic expression (or tuple/set of expressions) $\varphi$.

A typed **planning domain** $D = \langle T, P, A, M \rangle$ consists of:

- a set $T$ of types;
- a set $P$ of predicate symbols;
- a set $A$ of action symbols;
- an action model $M$ specifying the predicates in the preconditions, add and delete effects of each action schema.

The **types** in $T$ are organized into a tree (or hierarchy). We say that a type $t'$ **subsumes** type $t$ iff $t'$ is either $t$ or an ancestor of $t$ in the tree. Each predicate symbol $p \in P$ (resp. action symbol $a \in A$) has a signature $\mathrm{sig}(p)$ (resp. $\mathrm{sig}(a)$), that is a vector $\vec{t}$ of types such that $|\vec{t}|$ is the arity of $p$ (resp. of $a$). Given a set $X$ of variables used as arguments of the predicates and action schemas, a **predicate** takes the form $p(\vec{x})$ where $p$ is a predicate symbol and $\vec{x} \in X^{\mathrm{arity}(p)}$. Similarly each **action schema** takes the form $a(\vec{x})$ where $a$ is an action symbol, and $\vec{x} \in X^{\mathrm{arity}(a)}$, with $\vec{x}_i \neq \vec{x}_j \; \forall i \neq j$. We say that predicate $p(\vec{y})$ is **relevant** to action schema $a(\vec{x})$ iff each variable in $\vec{y}$ matches a variable in $\vec{x}$ with an appropriate type: $\forall i \in \{1, \ldots, \mathrm{arity}(p)\} \; \exists j \in \{1, \ldots, \mathrm{arity}(a)\}$ such that $\vec{y}_i = \vec{x}_j$ and $\mathrm{sig}(p)_i$ subsumes $\mathrm{sig}(a)_j$. We write $R(a(\vec{x}))$ for the set of predicates that are relevant to $a(\vec{x})$.

Given the predicate and action symbols and their respective signatures, we want to learn an **action model** $M$ mapping each action schema $a(\vec{x})$, to a triple $M(a(\vec{x})) = \langle \mathrm{Pre}(a(\vec{x})), \mathrm{Add}(a(\vec{x})), \mathrm{Del}(a(\vec{x})) \rangle$ of sets of predicates representing its preconditions, add effects and delete effects. Action model $M$ must satisfy the following:

- the predicates in $\mathrm{Pre}(a(\vec{x}))$, $\mathrm{Add}(a(\vec{x}))$, and $\mathrm{Del}(a(\vec{x}))$ must be relevant to $a(\vec{x})$;
- add effects and preconditions cannot intersect, i.e., $\mathrm{Add}(a(\vec{x})) \cap \mathrm{Pre}(a(\vec{x})) = \emptyset$;
- only preconditions can be deleted, i.e., $\mathrm{Del}(a(\vec{x})) \subseteq \mathrm{Pre}(a(\vec{x}))$. We borrow from SAS+ terminology and call preconditions that are not deleted prevail conditions (Bäckström and Nebel 1995).

Given $P$ and $A$, we write $\mathcal{M}(P, A)$ for the set of action models that satisfy those constraints.

A **planning instance** $I = \langle O, D \rangle$ consists of a set of objects $O$ and a planning domain $D$. Each **object** $o \in O$ is associated with a leaf type $\mathrm{type}(o) \in T$ of the type hierarchy. A **proposition** $p(\vec{o})$ with $p \in P$, $\vec{o} \in O^{\mathrm{arity}(p)}$, and such that $\mathrm{sig}(p)_i$ subsumes $\mathrm{type}(\vec{o}_i)$ for all $i \in \{1, \ldots, \mathrm{arity}(p)\}$, is a ground instance of a predicate $p(\vec{x})$ for some substitution $\sigma$ such that $p(\vec{x})[\sigma] = p(\vec{o})$. Similarly, an **action** $a(\vec{o})$ with $a \in A$, $\vec{o} \in O^{\mathrm{arity}(a)}$, and such that $\mathrm{sig}(a)_i$ subsumes $\mathrm{type}(\vec{o}_i)$ for all $i \in \{1, \ldots, \mathrm{arity}(a)\}$, is a ground instance of an action schema $a(\vec{x})$ for some substitution $\sigma$ such that $a(\vec{x})[\sigma] = a(\vec{o})$, and its action model is $M(a(\vec{x}))[\sigma]$. We write $P_I$ for the set of propositions, $A_I$ for the set of actions, and $S = 2^{P_I}$ for the set of **states** of the planning instance.

Let $I$ be a planning instance, $s \in S$ be a state, $a \in A_I$ be an action such that $M(a) = \langle \mathrm{Pre}(a), \mathrm{Add}(a), \mathrm{Del}(a) \rangle$. We say that $a$ is **applicable** in $s$ iff $\mathrm{Pre}(a) \subseteq s$. The result of applying $a$ in $s$ is the **successor** state $res(s, a) = (s \setminus \mathrm{Del}(a)) \cup \mathrm{Add}(a)$. An **execution trace** for planning instance $I$ is a sequence alternating between states and actions: $e = s_1 \rightarrow a_1 \rightarrow \ldots \rightarrow s_{|e|} \rightarrow a_{|e|} \rightarrow s_{|e|+1}$. We refer to $s_1$ as the **initial state** and $s_{|e|+1}$ as the **final state** of the trace. Trace $e$ is **consistent** with an action model $M$ if and only if, according to $M$, for all $i \in 1, \ldots, |e|$, $a_i$ is applicable in $s_i$ and $res(s_i, a_i) = s_{i+1}$. In the following, we write $\mathcal{E}_M^k$ for the set of execution traces of length $k$ that are consistent with action model $M$.

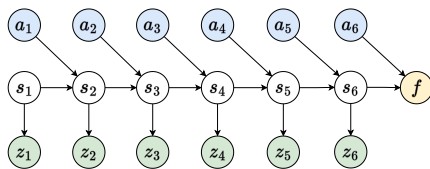

Figure 1: Observations in a visual trace $obs_e$ compared to the ground truth trace $e$. Symbol $s$ denotes states, $f$ is the final state, and $z$ denotes images. Shaded nodes are observed.

## 4 Problem Formulation

We now formalize the problem of learning action models from visual traces that we aim to solve. For an execution trace $e$, we only observe a **visual trace**, which is a sequence alternating images and actions: $obs_e = z_1 \rightarrow a_1 \rightarrow \ldots \rightarrow z_{|e|} \rightarrow a_{|e|} \rightarrow f$ where the $a_i$ represent fully observable actions and $f$ is the observed final state. Fig. 1 illustrates our observations in a 6-step visual trace compared to the ground truth execution trace. Note that for a given ground truth trace $e$ there can be multiple $obs_e$.

Given a planning instance $I$, a **probabilistic state vector** $\boldsymbol{ps} \in [0,1]^{|P_I|}$ for a state $s$ is a vector listing all propositions of the planning instance and their probabilities of being true in $s$. Such a vector can be estimated from a state image using a neural network. A probabilistic prediction of an execution trace $e$ from its visual trace observation $obs_e$ is a sequence alternating between probabilistic state vectors and actions: $pred_{obs_e} = \boldsymbol{ps_1} \rightarrow a_1 \rightarrow \ldots \rightarrow \boldsymbol{ps_{|e|}} \rightarrow a_{|e|} \rightarrow f$, where the predictions are made by a neural network with parameters $\theta$. The probability of state $s_i$, assuming independence of propositions, is

$$\Pr(s_i \mid z_i; \theta) = \prod_{p_j \in s_i} \boldsymbol{ps_{i_j}} \prod_{p_j \notin s_i} (1 - \boldsymbol{ps_{i_j}}). \quad (1)$$

The probability of trace $e$ given visual observation $obs_e$ is then

$$\Pr(e \mid obs_e; \theta) = \prod_{i=1}^{|e|} \left( \prod_{p_j \in s_i} \boldsymbol{ps_{i_j}} \prod_{p_j \notin s_i} (1 - \boldsymbol{ps_{i_j}}) \right). \quad (2)$$

In this work, we assume that we are given a set $\{obs_j\}_{j=1}^n$ of visual traces for a planning instance $I^M = \langle O, D^M \rangle$ with $D^M = \langle T, P, A, M \rangle$ for some unknown action model $M \in \mathcal{M}(P, A)$. Ideally, we wish to jointly choose an action model $M$ and neural network parameters $\theta$ that maximizes the log-likelihood of the visual traces being the observations of traces that are consistent with the action model,

$$\ell(M, \theta) = \sum_{j=1}^n \log \left( \sum_{e \in \mathcal{E}_M^{|obs_j|}} \Pr(e \mid obs_j; \theta) \right). \quad (3)$$

The requirement of having fully observable final states is to ensure that the learned results (both the state predictions and the action model) are human readable. It also ensures that we avoid degenerate solutions, e.g., empty sets for Pre,

Add and Del of all actions. However, we do not include images for final states because having both the image and label for the same state would effectively make learning to predict states fully supervised.

## 5 Probabilistic Action Model Network

Directly maximizing $\ell(M, \theta)$ is hard because it is intractable to compute the sum over all $e \in \mathcal{E}_M^{|obs_j|}$. Instead we relax the problem by modifying the successor state operator $res$ to be probabilistic and compute the expected next probabilistic state vector at each time step as $\widehat{\boldsymbol{ps}}_{t+1} = res(\boldsymbol{ps}_t, a_t)$. We then solve for,

$$\underset{\theta, M \in \mathcal{M}(P, A)}{\mathrm{argmin}} \sum_{j=1}^n \sum_{t=1}^{|obs_j|} \|\widehat{\boldsymbol{ps}}_{j,t+1} - \boldsymbol{ps}_{j,t+1}\|_2^2 + \mathcal{L}(a_t, \boldsymbol{ps}_{j,t})$$

where the final probabilistic state $\boldsymbol{ps}_{j,|obs_j|+1}$ is determined from $f$ and all other $\boldsymbol{ps}_{j,t}$ are estimated from $z_t$. Here we have added an additional term $\mathcal{L}(a_t, \boldsymbol{ps}_{j,t})$ to ensure that the observed action $a_t$ is applicable in the probabilistic state $\boldsymbol{ps}_{j,t}$ at step $t$. We provide further details in Section 6.

The above relaxation requires a way to compute probabilistic preconditions and effects of actions. However, since action models are defined symbolically, the above optimization problem is difficult to solve. Therefore, we relax the action model $M$ to a **probabilistic action model** with outputs interpreted as probabilities, so that our objective becomes fully differentiable with respect to $\theta$ and $M$ and amenable to standard back-propagation techniques.

**Definition 1** *A* Probabilistic Action Model *(PAM) is defined as a tuple of three functions $\langle pre, add, del \rangle$, where for an action schema $a(\vec{x})$ and a predicate $p(\vec{y})$ relevant to $a(\vec{x})$, $pre(a(\vec{x}), p(\vec{y}))$, $add(a(\vec{x}), p(\vec{y}))$, and $del(a(\vec{x}), p(\vec{y}))$ are probabilities of $p(\vec{y})$ being a precondition, an add effect, or a delete effect of $a(\vec{x})$.*

### PAM Cases

As discussed in the preliminaries, we assume that for any action model, add effects and preconditions cannot intersect, and only preconditions can be deleted. Consequently, we can enumerate all the possible relationships between a predicate $p(\vec{y})$ and an action schema $a(\vec{x})$ and determine whether they satisfy the above constraints. For any pair $(a(\vec{x}), p(\vec{y}))$ such that $p(\vec{y}) \in R(a(\vec{x}))$, there are four mutually exclusive cases that an action model can define:

- **Case 1:** $p(\vec{y})$ is not involved in the description of $a(\vec{x})$.
- **Case 2:** $p(\vec{y})$ is only an add effect of $a(\vec{x})$.
- **Case 3:** $p(\vec{y})$ is only a precondition of $a(\vec{x})$.
- **Case 4:** $p(\vec{y})$ is both a precondition and a delete effect, but not an add effect of $a(\vec{x})$.

Therefore, we can consider the task of learning action models as that of classifying the four cases for each pair of relevant action schema and predicate. A PAM gives a 4-vector $\overrightarrow{pr}_{a(\vec{x}), p(\vec{y})}$ for each pair of $a(\vec{x})$ and $p(\vec{y})$, where $p(\vec{y}) \in R(a(\vec{x}))$, which represents a probability distribution

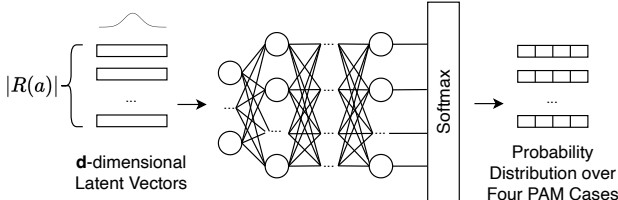

**d-dimensional Latent Vectors** · Softmax · **Probability Distribution over Four PAM Cases**

Figure 2: PAM network structure for an action symbol. The number of input neurons is $d$ and the PAM network is applied to each relevant predicate, where the output dimension is four. There is a batch of $|R(A)|$ $d$-dimensional latent vectors for each action symbol. The latent vectors are randomly drawn from a stand normal distribution and fixed during training.

over the four discrete cases. These probability distributions can be directly decoded into the functions $pre$, $add$ and $del$:

$$pre(a(\vec{x}), p(\vec{y})) = \overrightarrow{pr}_{a(\vec{x}),p(\vec{y})} \cdot (0,0,1,1) \quad (4)$$

$$add(a(\vec{x}), p(\vec{y})) = \overrightarrow{pr}_{a(\vec{x}),p(\vec{y})} \cdot (0,1,0,0) \quad (5)$$

$$del(a(\vec{x}), p(\vec{y})) = \overrightarrow{pr}_{a(\vec{x}),p(\vec{y})} \cdot (0,0,0,1) \quad (6)$$

These expressions can be interpreted as summing the probabilities over the respective compatible cases. For instance, the probability of $p(\vec{y})$ being a precondition of $a(\vec{x})$ is the sum of the probabilities of cases 3 and 4.

**PAM Network**

Directly learning the probability distributions for a PAM is highly non-convex and, therefore, very challenging. One method to mitigate this issue is overparameterization, which is often used to make optimisation problems smoother and to help the model converge to the global minimum (Du et al. 2019). Since our goal is to learn a discrete probability distribution, we also leverage the fact that, through a sufficiently complex function such as a neural network, any arbitrary distribution can be generated from a set of samples drawn from a Gaussian distribution (Doersch 2021). Combining these two ideas, for each action schema, we overparameterize its PAM into a PAM Network, with the inputs being a set of latent vectors drawn from a standard Gaussian — one for each of its relevant predicates. Subsequently, we create a multi-layer perceptron (MLP) with an output size of four, followed by a softmax layer to map from latent vectors to distributions over the four PAM cases. The dimensionality $d$ of the latent vectors is an empirically determined hyperparameter. The PAM network architecture is depicted in Fig. 2.

There are potentially infinitely many action schemas with different variable arguments for each action symbol. However, these schemas all share the same action model up to variable substitutions. Hence we only need the cardinality of the set of relevant predicates $R(a(\vec{x}))$ to initialize the PAM network for $a(\vec{x})$. This cardinality can be efficiently computed using action and predicate symbol signatures as explained below. As a result, we can confidently base our reasoning on a finite number of symbols without needing to enumerate an infinite number of predicates and action

schemas. We only need to construct one PAM network for each action symbol $a \in A$.

Given a predicate symbol $p \in P$ and its signature $\mathrm{sig}(p)$, we can easily count how many variables of each type $t \in T$ it requires. Let this be $\mathrm{count}(t, \mathrm{sig}(p))$. Similarly, given an action symbol $a \in A$ and its signature $\mathrm{sig}(a)$, we can count how many variables of a type **subsumed by** $t$ it provides. Let this be $\mathrm{subcount}(t, \mathrm{sig}(a))$. For each pair of symbols $a$ and $p$, if there exists a type $t$ such that $\mathrm{count}(t, \mathrm{sig}(p)) \neq 0$ and $\mathrm{subcount}(t, \mathrm{sig}(a)) = 0$, then we can infer that predicates with symbol $p$ are irrelevant to action schemas with symbol $a$. Otherwise, we can compute the desired cardinality as:

$$|R(a)| = \sum_{p \in P} \prod_{t \in T} \mathrm{subcount}(t, \mathrm{sig}(a))^{\mathrm{count}(t, \mathrm{sig}(p))} \quad (7)$$

Note that the construction of PAM networks is not based on objects. Therefore, PAM networks are decoupled from planning instances and are transferable within a domain.

We decode PAM network outputs using Eq. 4–6 to obtain PAMs for action symbols. For an action symbol $a$, the decoding results in three vectors $\boldsymbol{pre}_a$, $\boldsymbol{add}_a$, and $\boldsymbol{del}_a$, each of length $|R(a)|$, representing the lifted preconditions, add effects, and delete effects for $a$. These values are mapped to the corresponding positions of $|P_I|$-length vectors for generating propositional precondition and effect vectors for grounded actions, as will be discussed next.

## 6 ROSAME

Based on PAM networks, we create a neuro-symbolic model, named ROSAME, in order to compute the probabilistic preconditions and effects of actions. Fig. 3 shows the architecture of ROSAME, along with its inference on an action $a_t$ at the t-th step within a trace for a planning instance $I$. We compute three vectors $\boldsymbol{pre}_{a_t}$, $\boldsymbol{add}_{a_t}$, and $\boldsymbol{del}_{a_t} \in [0,1]^{|P_I|}$, where the $j$-th value represents the probability of proposition $p_j$ being the precondition, add or delete effect of action $a_t$, respectively. If a proposition is relevant[1] to $a_t$, these probabilities are determined by the corresponding PAM for $a$. Otherwise, the action cannot affect or be affected by the proposition, and all three values are set to zero.

We maintain an ordering on all the propositions in the planning instance. After grounding, we record the indices of relevant propositions within the ordered list of all propositions for each action, as well as the mapping from each action to its action symbol. Therefore, given an action $a_t$, we can lift the action to its symbol $a$ and utilize the corresponding PAM Network to compute the lifted preconditions and effects $\boldsymbol{pre}_a$, $\boldsymbol{add}_a$, $\boldsymbol{del}_a$ for $a$. Subsequently, we map values in these vectors, each of length $|R(a)|$, to the relevant indices in the full $|P_I|$-length vectors $\boldsymbol{pre}_{a_t}$, $\boldsymbol{add}_{a_t}$, and $\boldsymbol{del}_{a_t}$ for the propositional preconditions and effects of the grounded action $a_t$.

After training, we create one action schema $a(\vec{x})$ for each action symbol $a$ and extract its action model $M(a(\vec{x}))$. This

---

[1]The notion of relevance straightforwardly transfers from action schemas and predicates to actions and propositions obtained by applying the same substitution to the variables.

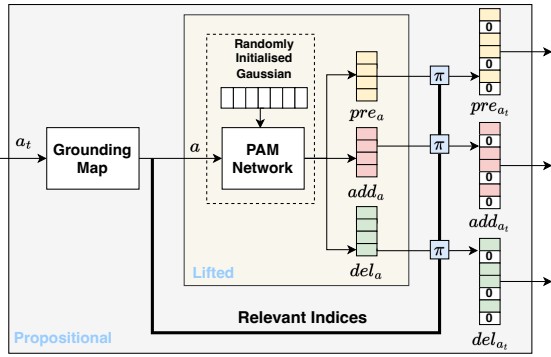

Figure 3: ROSAME architecture. The projection operation $\pi$ maps the output of the PAM network to relevant indices in vectors of length $|P_I|$. Indices not mapped take value zero.

is done by computing an ordered list of relevant predicates for $a(\vec{x})$. Here we map each output vector from the PAM Network for $a$ to the relevant predicates in order. Then, based on the classification results over PAM cases, we add the predicates to the sets in the action model $M(a(\vec{x}))$.

ROSAME is independent of planning instances. The learnable parameters relating to the action model are fully contained within the PAM network, structured solely based on domain knowledge. The only difference between two planning instances in the same domain is the set of relevant proposition indices for each grounded action. A new planning instance only requires rerunning the grounding process on a new list of objects without any changes to the PAM network. As a result of learning a lifted action model, ROSAME is able to transfer to other problem instances. Specifically, we can efficiently train ROSAME on a small instance and then apply it to a much larger instance within the same domain without retraining.

## Training Loss

We now detail the loss function used to train ROSAME. Given an action model $M$ and an execution trace $e$ that is consistent with it, by the definitions of consistency and of the successor state operator $res$, $\forall p \in P_I, \forall t \in \{1, \ldots, |e|\}$:

$$
\begin{aligned}
p \in s_{t+1} &\iff \big( p \in s_t \wedge \neg(p \in \mathrm{Del}(a_t)) \big) \vee \\
& \quad \big( \neg(p \in s_t) \wedge p \in \mathrm{Add}(a_t) \big), \quad (8) \\
p \in \mathrm{Pre}(a_t) &\implies p \in s_t
\end{aligned}
$$

The first formula states that a proposition $p$ holds in state $s_{t+1}$ if and only if either $p$ holds in state $s_t$ and is not deleted by action $a_t$, or $p$ does not hold in state $s_t$ but is added by action $a_t$. Note that the first condition includes the case where $p$ both holds in state $s_t$ and is an add effect of $a_t$, because being an add effect implies not being a delete effect, as per our assumptions in Section 3. The second formula in Eq. 8 states that if a proposition $p$ is in the precondition of an action $a_t$, and we have observed that $a_t$ was applied at step $t$, then $p$ must hold in state $s_t$ before action $a_t$ is executed.

For any step $t$ in $1, \ldots, |obs|$, we can use ROSAME to infer the next state $\widehat{\boldsymbol{ps}}_{t+1}$ by applying the the product logic

rules (Hájek, Godo, and Esteva 1996) to the first formula in Eq. 8. For all $p \in P_I$, $\mathrm{Pr}(p \in s_{t+1}) = \mathrm{Pr}(p \in s_t) \times (1 - \mathrm{Pr}(p \in \mathrm{Del}(a_t))) + (1 - \mathrm{Pr}(p \in s_t)) \times \mathrm{Pr}(p \in \mathrm{Add}(a_t))$, hence we have:

$$
\widehat{\boldsymbol{ps}}_{t+1} = \boldsymbol{ps}_t \times (1 - \boldsymbol{del}_{a_t}) + (1 - \boldsymbol{ps}_t) \times \boldsymbol{add}_{a_t} \quad (9)
$$

Note that here, we can translate the logical disjunction into the summation of two probabilities because these two cases are mutually exclusive. A PAM that is consistent with a probabilistic prediction of a trace $pred$ should infer next-step states close to $pred$. Hence, we compute the mean square error (MSE) between $\widehat{\boldsymbol{ps}}_{t+1}$ and $\boldsymbol{ps}_{t+1}$ at each step $t$ in the trace, where $\boldsymbol{ps}_{|pred|+1}$ is determined by the fully observable final state.

To calculate the probability of $a_t$ being applicable in state $s_t$, we rephrase the second formula in Eq. 8 as a combination of negation and conjunction: $\neg\big( p \in \mathrm{Pre}(a_t) \wedge \neg(p \in s_t) \big)$. This probability can be computed as $1 - \boldsymbol{pre}_{a_t} \times (1 - \boldsymbol{ps}_t)$. There should be a high probability of each action being applicable at each step. We use an MSE between the computed probability and an all-ones vector $\boldsymbol{1}$ to reflect this fact.

**Prevail conditions** A prevail condition of an action is a precondition that is not deleted by the action (Bäckström and Nebel 1995). Prevail conditions correspond to PAM case 3. In an execution trace, the prevail condition holds both before and after the execution of the action. However, this information alone is not sufficient to distinguish PAM case 3 from PAM cases 1 and 2, where the proposition is not involved in the description of the action or it serves as an add effect. Such confusion causes indistinguishability among models.

We introduce an additional prior bias to address the indistinguishability problem. We assume that a relevant predicate is a precondition of an action schema unless evidence from data contradicts this assumption. Therefore, we give preference to the model with the prevail condition. This prior bias not only increases the likelihood of recovering prevail conditions but also results in a more conservative action model, which can be valuable in safety-critical situations. In practice, we introduce this prior bias using a loss term for each action, computed as the MSE between $\boldsymbol{pre}_{a_t}$ and an all-ones vector $\boldsymbol{1}$ at each step.

Given a set $\{pred_j\}_{j=1}^n$ of predictions for traces, the loss used to train ROSAME is

$$
\ell(\theta, M) = \sum_{j=1}^{n} \sum_{t=1}^{|pred_j|} \overbrace{\mathrm{MSE}(\widehat{\boldsymbol{ps}}_{t+1}, \boldsymbol{ps}_{t+1})}^{Loss_{pred}} +
$$
$$
\underbrace{\mathrm{MSE}(\boldsymbol{pre}_{a_t} \times (1 - \boldsymbol{ps}_t), \boldsymbol{0})}_{Loss_{app}} + \lambda \cdot \underbrace{\mathrm{MSE}(\boldsymbol{pre}_{a_t}, \boldsymbol{1})}_{Loss_{bias}}
$$

where $\lambda < 1$ is an empirically determined value that reflects the influence scale of the prior bias.

## ROSAME-I

We propose ROSAME-I (ROSAME from Images), an end-to-end framework that combines ROSAME with a deep learning computer vision (CV) model to learn action models from visual traces. Fig. 4 illustrates ROSAME-I's learning process from a single visual trace. At the $t$-th step, the

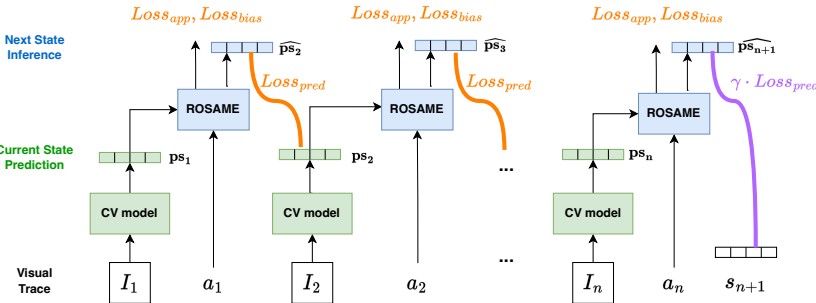

Figure 4: Learning action model on a visual trace with ROSAME-I.

CV model predicts $ps_t$ from the observed state image. Subsequently, ROSAME uses the CV model's prediction $ps_t$ and the action $a_t$ to infer the next state $\widehat{ps}_{t+1}$ and calculates the action's applicability loss $Loss_{app}$ and the prior bias $Loss_{bias}$. After that, we compare ROSAME's inference $\widehat{ps}_{t+1}$ with the CV model's prediction for the next step, $ps_{t+1}$, resulting in the prediction loss $Loss_{pred}$.

We assume that we have access to the ground truth labels for the final state as supervision. To emphasize our focus on prediction consistency with the supervision, we introduce a hyperparameter $\gamma \geq 1$ to scale the prediction loss at the last step. Intuitively, this hyperparameter controls the balance between ROSAME-I correctly predicting the final state and making coherent predictions for the previous states while adhering to the logical constraints of action models.

## 7 Experiments

### Data and Environment

We create two visual representations to evaluate ROSAME-I. First, we create state images using digit and letter figures from the MNIST (Deng 2012) and EMNIST (Cohen et al. 2017) datasets to represent objects and backgrounds, arranging them in grids. We refer to this as the grid world representation. With this representation, we can efficiently construct images for states and automatically generate visual traces from simulations, allowing us to develop and test the end-to-end nature of our method in a controlled setting. Fig. 5 (left) provides examples of the grid world representations for the three domains we consider, Blocksworld, Gripper, and Logistics, along with the state they represent. Note that changing the order and positions of block towers in the Blockworld domain, the positions of balls within the same room in the Gripper domain, or the object locations within each $3 \times 3$ grid in the Logistics domain does not change the underlying state that the image represents.

For the Blockworld domain, we use an off-the-shelf random problem generator (Slaney and Thiébaux 2001) to create the initial states. Traces are generated from these initial states by selecting random applicable actions at each step. For the Gripper and Logistics domains, we utilize the trace generation component from the MACQ framework (Callanan et al. 2022) to create long random traces, which are subsequently divided into shorter traces as required. Specifically, for each trace generated in all three domains,

we randomly select digit and letter figures. These figures remain consistent within an individual trace but are re-selected randomly for each new trace giving diverse representations.

Next, we create a synthesized representation for the Blockworld, Tower of Hanoi, and 8-puzzle domains. We utilize the PDDLGym framework (Silver and Chitnis 2020) to construct reinforcement learning environments in which we generate long traces through random exploration and then cut them into traces of the required length. This synthesized representation is more holistic and natural for humans to recognize. Example images for this representation are displayed on the right of Fig. 5.

ROSAME-I is implemented using PyTorch. The code will be publicly released upon publication of the paper. We train and test ROSAME-I on the Google Colab Platform, with 83.5 GB RAM and a single A100 40GB GPU.

### ROSAME Performance

In addition to visual traces, we collect fully observable traces to assess ROSAME as a stand-alone tool. We sample traces from the same domains and problem instances used for creating the grid world representations. We train ROSAME on the dataset in a fully supervised manner. We set the PAM Network latent dimension $z$ to be 128 and the prior bias scale $\lambda$ to be 0.2. We train the models for 100 epochs using the Adam optimiser with a learning rate of 0.001.

| | # Traces | # Steps | # States | Error |
|---|---|---|---|---|
| Blockworld | 10 | 10 | 100 | 0 |
| Gripper | 10 | 10 | 100 | 0 |
| Logistics | 10 | 10 | 100 | 0 |

Table 1: Data and performance of ROSAME.

As we treat the action model learning task as classifying the four PAM cases for relevant action schemas and predicates, we establish the Error metric for the learned models as the count of misclassifications compared to the ground truth models written by human. Results in Tab. 1 demonstrate that ROSAME perfectly recovers the ground truth models for the three domains with a limited amount of training data. The training process completes in less than a minute without GPU acceleration, highlighting the efficiency of our model.

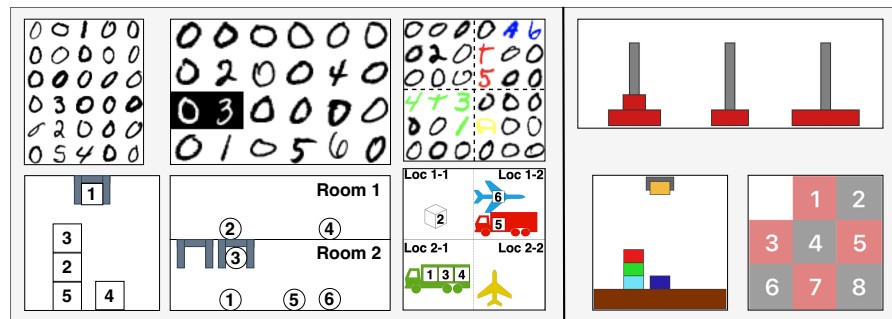

Figure 5: Domains and visual representations. Left: Grid world representations (top) and corresponding hand-drawn states (bottom) for Blockworld, Gripper, and Logistics domains. Digits 0 represent backgrounds, while other digits represent objects, including blocks, balls, and packages. In the Gripper domain, flipped colors represent the two grippers. In the Logistics domain, letters 'A' represent airplanes, and letters 'T' represent trucks; the same color indicates a package is carried by a vehicle. Right: Synthesized representations for Blockworld, Tower of Hanoi, and 8-puzzle domains.

| | # Grid Classes | $|P_I|$ | # Traces | # Steps | #States | #Epochs | Learning Rate | Error | Acc |
|---|---|---|---|---|---|---|---|---|---|
| Blockworld (grid world) | 6 | 36 | 800 | 6 | 4800 | 200 | Grid CNN: $10^{-5}$ MLP: $10^{-3}$ ROSAME: $10^{-3}$ | 0 | 97.51% |
| Gripper | 14 | 28 | 1000 | 5 | 5000 | 100 | | 0 | 90.54% |
| Logistics | 35 | 72 | 2500 | 10 | 25000 | 150 | | 0 | 96.41% |
| Blockworld (synthesised) | N/A | 36 | 100 | 10 | 1000 | 50 | $10^{-3}$ | 0 | 95.26% |
| Tower of Hanoi | N/A | 91 | 70 | 5 | 350 | 70 | | 1 | 99.64% |
| 8-puzzles | N/A | 330 | 300 | 5 | 1500 | 100 | | 4 | 99.67% |

Table 2: Evaluation result for ROSAME-I.

## ROSAME-I Performance

We combine ROSAME with a customized CV model for the grid world representation. The CV model first applies to each grid image a CNN classifier whose architecture is adopted from LeNet (LeCun et al. 1998). The CNN outputs for each grid are concatenated and processed through a multi-layer perceptron (MLP) to predict the truth values of state propositions. For the synthesized images, we utilize a ResNet-18 (He et al. 2016) and again replace the last fully-connected layer with an MLP to predict state propositions. We set the PAM Network latent dimension $z$ to be 128, the prior bias scale $\lambda$ to be 0.2 (except for the 8-puzzle domain, where we use a $\lambda$ of 0.4), and the supervision bias $\gamma$ to be 10. We use an Adam optimizer with $\beta = (0.9, 0.999)$.

We evaluate the quality of the learned action model with the Error metric defined above. For the CV model, we compute the proposition prediction accuracy with a threshold of 0.5. Tab. 2 presents the evaluation results for ROSAME-I. Our method recovers almost perfect action models across different domains and with various visual representations. Simultaneously, the CV models within ROSAME-I learn to predict states from images accurately without incurring additional labelling or training costs.

We reserve 10% of the traces collected as test traces for the grid world representation. For the synthesized representation, we create and reserve another dataset with 100 traces for testing. Note that we create our datasets by generating random traces. Although we reserve the test traces from training, some test state images and traces may still appear in

the training set due to the possibility of duplications within our entire dataset. However, we consider this less of a problem for evaluating the CV models because we never provide direct supervision for any state images. For ROSAME-I to learn to predict states, it must progressively transfer supervision from back to front through ROSAME. Therefore, the high accuracy of CV model predictions demonstrates that ROSAME can effectively reason with the supervision signal and convey it to the CV model for learning.

## Errors in Tower of Hanoi and 8-puzzles

Upon closer examination of the errors appearing in Tab. 2, we discover that the discrepancies are due to additional prevail conditions. Tab. 3 shows the model learned by ROSAME-I for the Tower of Hanoi domain. The learned model has an additional prevail condition that requires a disc to be placed on top of a larger disc (or peg) before it can be moved. It is worth noting that this condition trivially holds for any valid states in the Tower of Hanoi domain. Therefore, the learned model is actually correct despite being different from the ground truth. In fact, one might argue that the learned model is better than the human-crafted one, as the additional precondition can serve as a safety check, resulting in safer behaviour for some invalid problem instances.

It is not surprising that ROSAME-I learns this additional prevail condition. The model is only trained on valid traces, where the condition consistently holds before the execution of move actions. With the prior bias we introduced, ROSAME-I is encouraged to include this condition as a pre-

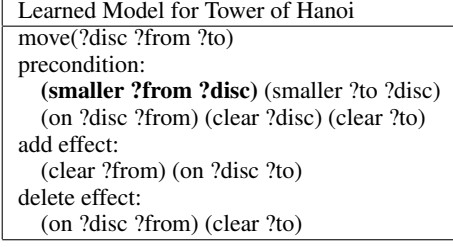

| Learned Model for Tower of Hanoi |
| --- |
| move(?disc ?from ?to) |
| precondition: |
|    **(smaller ?from ?disc)** (smaller ?to ?disc) |
|    (on ?disc ?from) (clear ?disc) (clear ?to) |
| add effect: |
|    (clear ?from) (on ?disc ?to) |
| delete effect: |
|    (on ?disc ?from) (clear ?to) |

Table 3: The action model learned by ROSAME-I for Tower of Hanoi. The predicate in bold does not appear in the ground truth model written by human.

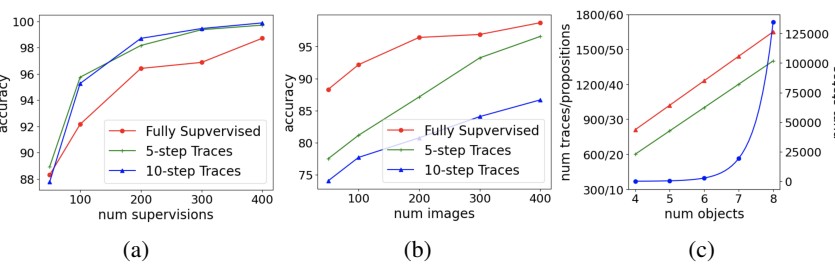

(a)         (b)         (c)

Figure 6: Data efficiency of the CV model in ROSAME-I (a,b) and scalability with respect to problem size (c).

condition unless there is sufficient counter-evidence in the data. This mechanism also aids ROSAME-I to recover the other three prevail conditions in the Tower of Hanoi domain.

In the preconditions of the move actions for the 8-puzzle domain, the learned model additionally recovers relationships between the target position and the original position that are symmetrically reversed compared to their relationships in the ground truth model, e.g., including both $\mathrm{dec}(?by, ?py)$ and $\mathrm{inc}(?py, ?by)$ as preconditions of moveup, resulting in correct but redundant prevail conditions.

### Data Efficiency

We hypothesize that the presence of ROSAME enhances the data efficiency of the CV model within ROSAME-I because ROSAME can infer with the evolving action model and propagate a single supervision to multiple state images for the CV model to learn from. To test this hypothesis, we assess the performance of the CV model extracted from two ROSAME-I models trained in the Blockworld domain with the synthesized representation, using traces of 5-step and 10-step lengths, respectively. In each case, there is one supervision in each trace. In contrast, we train a separate CV model with the same architecture using state images with fully supervised proposition labels. We then compare the prediction accuracy among the models. As shown in Fig. 6a, it is evident that with the same amount of supervision, the CV models extracted from ROSAME-I consistently outperform the CV model trained in isolation. This result confirms that data efficiency increases with the assistance of ROSAME.

In contrast, Fig. 6b displays the model's performance with respect to the number of state images observed by the model. It is clear that models extracted from ROSAME-I, trained on traces where multiple images share a single goal state supervision, suffer performance loss because the level of supervision decreases. The longer the trace, the fewer supervisions are available for the same number of states, leading to a decline in model performance.

### Scalability with respect to Problem Size

Using the Logistics domain, we examine the scalability of our method with respect to problem size. We generate problem instances that consist of two trucks, one plane, and two cities, each with two locations, and use the grid world representation. We vary the number of packages and determine the problem's size by counting the total number of objects (comprising only trucks, planes, and packages since cities and locations are static).

For each problem with a different size, we increase the number of traces for training until ROSAME-I recovers the ground truth model, resulting in the green line in Fig. 6c. The figure also shows number of propositions (in red), and the number of different states (in blue), which grow linearly and exponentially, respectively, as a function of the problem size. As is evident from the figure, the amount of data required for training ROSAME-I aligns with the proposition space, avoiding the combinatorial nature and the exponential growth of the problem state space. This result demonstrates the satisfying scalability of our method.

### Ablation Studies and Shortcuts

We conducted ablation studies to examine the effects of introducing prior bias and overparameterizing PAMs with neural networks, as opposed to directly learning the probability distributions over PAM cases. We also analyzed reasoning shortcuts, a specific problem associated with neurosymbolic methods we encountered in our research. We have included these sections in the supplementary materials.

## 8 Conclusion and Future Work

This paper presents ROSAME-I, an end-to-end neurosymbolic model that learns lifted action models from visual traces. This learning process is guided by the given types, predicates, action symbols, their signatures, and the objects within the planning instance. We evaluated ROSAME-I across various domains employing different visual representations and achieved high-quality action models. While in this paper, we perform one-step inferences on successive states using ROSAME, replacing it with multi-step inference on future states could offer stronger regularization and enhance the learning task. As we progress towards our ultimate goal of learning action models from plan demonstration videos with minimal supervision, an intermediate step could involve reducing the need for action labelling in visual traces through a deep learning action classifier. To enable fully automated learning from videos, we will also need to develop methods for segmenting continuous video streams into discrete states and actions.

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
