# OpenReview forum: "Neuro-symbolic Learning of Lifted Action Models from Visual Traces"
_icaps-conference.org/ICAPS/2024/Conference — ICAPS 2024_

### Official Review · Reviewer_fS5z · 2024-01-09

**Significance And Importance:** 2
**Soundness:** 3
**Novelty:** 2
**Clarity:** 4
**Overall Evaluation:** 2
**Confidence:** 4

**Weaknesses:**

1: Minor weaknesses that are easily fixable.

**Contributions Of The Paper:**

The paper proposes ROSAME-I, a method for learning lifted action models from visual plan traces. The proposed method is implemented by a neuro-symbolic architecture (trained end-to-end) that combines deep learning models adopted in Computer Vision (CV) with logical constraints of action models. Considering a set of visual traces of a specific domain, ROSAME-I jointly learns to predict symbolic states from state images and symbolic action models given the domain predicates, object types, action symbols with their signatures, and objects in the planning instances used for generating the visual traces. ROSAME-I provides high-quality action models in several planning domains with different visual representations.

**Ethical Considerations:**

(1) Not Applicable: The paper does not have any ethical considerations to address

**Nomination For Best Paper:**

No

**Questions For Authors:**

Q1. Could you provide insights on the importance of the hyperparameter gamma in Figure 4? For example, how ROSAME-I performance vary when gamma equals 1? How does the length of the traces relate to gamma?
Q2.  Do you assume that delete effects must be positive preconditions?
Q3. Does the data efficiency refer to the number of images or supervisions?


Post-rebuttal: I thank the authors for their exhaustive answers to my questions.

**Reproducibility:**

4: Authors promise to release code and domains (whichever apply).

**Strengths Of The Paper:**

1.	[Importance] I think the action model learning problem is very challenging (specially from visual plan traces) and its importance has been widely recognized by the planning community.
2.	[Novelty] Despite there are neuro-symbolic approaches (e.g. LatPlan) for learning action models from state images, they mainly focus on learning latent state representations. On the contrary, ROSAME-I learns human-readable state representations, as the authors highlighted also in Section “Related Work”. Moreover, ROSAME-I adopts a loss function that considers logical knowledge of action models and allows for an end-to-end training with CV deep learning models.
3.	[Generalization] The output of a PAM network can be used for specifying lifted action models, which makes PAM networks generalize over different planning instance of a specific domain.
4.	[Data efficiency] In the experimental evaluation, I found valuable that ROSAME improves the data efficiency of the considered deep learning models for learning symbolic state representation. However, if I understood correctly, the efficiency is improved in terms of number of supervisions, rather than number of images. For example, in Figure 6a, given the same number of 100 supervisions, the green and blue curves are the performance obtained with 500 and 1000 images, while the red curve have access to only 100 images.
I believe the paper is very well written and the experimental evaluation is convincing.

**Weaknesses Of The Paper:**

1.	[Propositions independence] I think the state propositions independence assumption, adopted in Equation (1), is not always reasonable. For example, in domain Blocksworld, on(x1, x2) being true implies that on(x1, x3) is false for every x3 != x2.
2.	[Full observability] As far as I understood, the approach requires the states to be fully observable. For example, there cannot be occluded areas in the state images.
3.	[Final state supervision] On the one hand, I think the assumption of having final state labels can be a limitation. On the other hand, I acknowledge it to be an (according to me acceptable) trade-off for learning human readable representations. Moreover, it is not clear to me how much the parameter gamma affects ROSAME-I performance, especially when the length of the traces increase.

Minor notes:
1.	If I understood correctly, you assume that delete effects are positive preconditions. If this is the case, such an assumption might not always be reasonable (e.g., domain Satellite).
2.	Given the high performance of ROSAME-I in the domains considered in the experimental evaluation, I think it would have been interesting to also highlight the possible weaknesses of ROSAME-I. For example, how the performance varies when the number of traces in Table 2 decreases, or the length of the traces increases.
3.	I think you assume that preconditions cannot be negative, but I was not able to find this assumption explicitly stated in the paper.
4.	There is a typo in the legend of Figures 6a and 6b.

---

> ### Author Rebuttal · Authors · 2024-01-28
>
> We plan to purchase an extra page to handle some of your comments and add additional results to our git repo.
>
> Q1. Gamma insight: We use an empirically determined gamma value of 10 across all experiments. While it is related to the trace length, we have observed a wide range of values is suitable for gamma. However a gamma value of 1 is often insufficient for accurately recovering the ground truth model. E.g., for Blocksworld (grid), we tested gamma=1,5,7,10,15,20; the CV model accuracy is 83.99%, 93.57%, 97.62%, 98.27%, 95.72%, 96.20% respectively, and the error drops from 16 (gamma=1), to 8 (gamma=5), to 0 (all other values).
>
> Q2. Whilst the paper assumes that delete effects must be positive preconditions, there are several potential ways to remove this assumption:
>
> - We can add a 5th PAM case, where a predicate functions as a delete effect without being a precondition. We would then decode this case accordingly.
>
> - In preliminary exploration, we modeled the probability of a predicate being a precondition, add, or delete effect separately. We used product logic to enforce constraints on impossible cases, such as ensuring that the probability of a predicate being both a precondition and an add effect approaches 0. This approach is more flexible and can address the proposed issue, but the current version that hardcodes the allowed cases makes model training easier and improves performance.
>
> The same considerations apply to negative preconditions (indeed we currently use the STRIPS fragment of PDDL).
>
> Q3. Data efficiency refers to the level of supervision. Your comment on Fig. 6(a) in the strength section is correct.
>
> Other questions:
>
> Propositions independence:  We agree that this assumption can sometimes be violated, hence Eq. 2 is an overestimate of the probability. Nevertheless, it serves to motivate the derivation of the loss function, which we then relax in Sec. 5, and still provides a useful training signal. Indeed, the CV model used for ROSAME-I does not enforce independence on outputs ps_t, nor does any other component of our model. We will clarify this when revising the paper.
>
> Full observability: We run our experiments using fully observable images. However, the CV model, being a neural network that makes probabilistic predictions, is not limited to fully observable images. The CV model always generates predictions for the propositions, although their quality may be affected by observability.
>
> Impact of the trace number/length: see FGQr/Q4.

---

### Official Review · Reviewer_NuY9 · 2024-01-21

**Significance And Importance:** 2
**Soundness:** 3
**Novelty:** 3
**Clarity:** 3
**Overall Evaluation:** 1
**Confidence:** 3

**Weaknesses:**

1: Minor weaknesses that are easily fixable.

**Contributions Of The Paper:**

The paper presents ROSAME, a differentiable neuro-symbolic action model learner that can learn PDDL models from images using minimal human input. To do so, the authors utilize annotated versions of the final states in a trace and utilize probabilistic action model (PAM) networks to minimize the loss between the vectorized representations of the final (and intermediate states). The state vector is simply an encoding of all possible propositions over the state. However, outputs from PAM networks only handle the cases that arise based on whether a lifted predicate appears as a part of the precon or add/delete lists. The restriction here is that predicates not belonging in the preconditions cannot be deleted.

The intuition is that when there is a vectorized representation of the state vector, product logic can be used to generate the next state using the PAM network. Thus, an end-to-end framework can be identified where both the state representation and action models can be learned simulataneously which forms the basis for ROSAME-I. To identify correct preconditions, the authors propose a pessimistic approach that biases the models to include predicates as a part of the precondition unless counter-examples are found.

The authors then provide an empirical evaluation and show that their approach can recover ground truth models.

**Ethical Considerations:**

(5) Excellent: The paper comprehensively addresses all of the applicable ethical considerations

**Nomination For Best Paper:**

No

**Questions For Authors:**

1. Could you please comment on the identified weaknesses? Particularly, is there any reason why no baselines were included?

2. How are lifted models extracted from the PAM networks? This particularly important detail is missing from the main paper and is only briefly mentioned.

Could you elaborate on line 330 onwards? "Then, based on the classification results over PAM cases, we add the predicates to the sets in the action model M (a(~x))."

Post rebuttal
===========
Thank you for clarifying my questions. If accepted, I hope the authors are able to improve the clarity of the paper w.r.t. the points in #2 and #3 that I;ve mentioned. I wish the authors all the best.

**Reproducibility:**

4: Authors promise to release code and domains (whichever apply).

**Strengths Of The Paper:**

1. The paper is clear and well-written
2. The end-end framework for learning vectorized state encodings and PDDL models from a dataset of traces with only the final state annotated is significant.

**Weaknesses Of The Paper:**

1. The input requirements are still quite high. The approach needs access to the symbols (predicates) and action signatures in order to work.
2. The empirical section is a bit hazy and some configurations are missing. For example, what are the sizes of problems in blocksworld etc that were used? How were the trace cuts of suitable length identified? Were they always from the initial state upto length L or were traces cut from states that were not the initial state?
3. It is not clear how lifted models are extracted from the propositional encoding. The authors describe it near line 330 but the details are not present.

---

> ### Author Rebuttal · Authors · 2024-01-28
>
> Q1. Baselines: Please see our reply to reviewer FGQr/Q5.
>
> Q2. Extraction of lifted models from the PAM networks: Please note that we know the correspondence between each latent vector in the PAM network and each predicate relevant to the action symbol; this knowledge is essential for mapping the lifted preconditions and effects to the propositional preconditions and effects in order. Therefore, we do not lift the model from the propositional encoding. Instead, to extract the model for an action symbol, we take the corresponding latent vector for each of its relevant predicates, pass it through the PAM network, and then add the predicate to the Pre, Add, and Del sets based on the classification result. We will clarify this.
>
> Other questions:
>
> Input requirements being high and access to predicate and action signatures: We agree that this is a limitation. Reducing the input requirements is an important future work. For instance, we plan to investigate possible symbol creation methods in conjunction with our approach.
>
> Problem sizes and trace cuts: Thank you for pointing this out. Certain details were omitted due to page limits. Regarding problem sizes, we use the sizes shown in Fig 5.: 5 blocks in Blocksworld, 6 balls, 2 rooms, and 2 grippers in Gripper, 6 packages, 2 cities (each with 2 locations), 2 trucks, and 2 planes in Logistics, and 4 disks in Hanoi. We will add this info to the text. We empirically determined a trace length that reflects the complexity of the domain and problem size -- either 5 or 10 steps (note there is a typo in the table: Blocksworld should be 10 steps). However, it's worth noting that for any practical purposes, one should collect as much data as possible. The method we used to cut the traces is described around line 440. When we say "cut", we generate a very long random trace from a single initial state and then divide it into many shorter traces of the selected length. This means that, except for the first trace, later traces started from states that were not the initial state. We do skip some states to avoid overlap between traces.
>
> Other comments:
>
> Restriction that only preconditions can be deleted: see reviewer fS5z/Q2.

---

### Official Review · Reviewer_FGQr · 2024-01-22

**Significance And Importance:** 3
**Soundness:** 3
**Novelty:** 3
**Clarity:** 3
**Overall Evaluation:** 2
**Confidence:** 3

**Weaknesses:**

1: Minor weaknesses that are easily fixable.

**Contributions Of The Paper:**

- This paper introduces ROSAME, a neurosymbolic framework to automatically learn action models from visual traces.
- The proposed framework has been combined with a CNN to create an end-to-end framework capable of automatically infer symbolic action models.
- Experimental results showcase that the proposed framework succeeds in generating symbollic actions, generating traces that are either equal or better to those manually proposed by humans

**Ethical Considerations:**

(4) Good: The paper adequately addresses most, but not all, of the applicable ethical considerations

**Nomination For Best Paper:**

Yes

**Questions For Authors:**

1. Since the method is fully unsupervised, how is it checked whether the obtained traces are correct?
2. Should actions be predefined before being mined from the image (i.e., as in image classification), or are they also detected solely from the image?
3. For the parameter lambda, which values should it have? How is its optimal value computed?
4. How does the number of traces available impact the final results?
5. Why does the proposal have not being compared with other similar approaches? It would have been interesting to see a comparison.

**Reproducibility:**

4: Authors promise to release code and domains (whichever apply).

**Strengths Of The Paper:**

- The contributions are clearly outlined and the framework is presented in a clear and easy to follow manner.
- All elements are clearly described and claims are supported by its corresponding equations and definitions.
- The performed experimentation on the performance of the model is sufficient and the presented results are quite remarkable.

**Weaknesses Of The Paper:**

- It misses comparison with other similar models
- Some minor technical definitions are missing.
- Figure 6 is a bit small and hard to read.

---

> ### Author Rebuttal · Authors · 2024-01-28
>
> Thanks for your insightful comments. We plan to purchase an extra page to handle some of them and add additional experiments to our git repo.
>
> Q1. Trace correctness: Our training data is generated from a ground truth model through simulation, hence we know that the traces are correct.
>
> Q2. Actions predefined or detected: Yes, the action signatures are known. We also observe the actions being performed in traces (see Problem Formulation p3). We will further clarify this in the introduction to remove any ambiguity.
>
> Q3. Lambda value: The hyperparameter is empirically determined. It is meant to bias the model towards including prevail conditions. We set lambda to 0.2 for all the domains except 8-puzzle, where lambda is set to 0.4 in recognition of the abundance of static predicates in this domain. We have run an ablation study on lambda -- see supplementary material. We will try to provide a more thorough study.
>
> Q4. Impact of the number/length of traces: As shown in Fig. 6(a), the CV model's prediction accuracy increases with the number of traces (where the number of supervision is the number of traces, i.e., one state label per trace). Moreover, we observed that ROSAME-I's performance improves with more traces and/or longer trace steps. E.g., for Blocksworld (grid), when we fix the trace length to be 10 and vary the number of traces to be 400, 600, 800, 1000, the CV model's accuracy increases (87.00%, 89.18, 98.27%, 98.49%) and the action model error drops (1, 1, 0, 0). When we fix the number of traces to be 800 and vary the trace length to be 3, 5, 7, 10, 12 (gamma fixed to be 10), the CV model's accuracy increases (93.13%, 94.36%, 95.88%, 98.27%, 98.52%) and the action model error decreases (1, 0, 0, 0, 0).
>
> Q5. Comparison with baselines:  We are not aware of any existing work that addresses a task that is sufficiently close to fairly compare and draw conclusions. Latplan is fully unsupervised and produces latent models. Other approaches do not train end-to-end from images. We therefore decided to instead use the fully supervised version of ROSAME as a baseline, and to focus on evaluating the performance of the integrated model on visual traces. We however welcome any suggestion of a more suitable baseline or suitable models to compare to, and would be happy to incorporate any relevant comparison in the final version. We will also release our code to ensure that all our results can be reproduced for future comparative analyses.

---

### Meta-Review · Area_Chair_cjXM · 2024-02-03

**Recommendation:** Accept (Oral)
**Confidence:** 4

**Metareview:**

This paper proposes ROSAME, a system which seeks to learn human-readable, lifted action models and state predictors from sequences of images.
The reviewers agree that the paper addresses a challenging problem and makes significant progress. All reviewers were satisfied with the author rebuttals.

**Ethical Considerations:**

(1) Not Applicable: The paper does not have any ethical considerations to address